# Transformation from crystalline precursor to perovskite in PbCl$_2$-derived MAPbI$_3$

Kevin H. Stone [1], Aryeh Gold-Parker[1,2], Vanessa L. Pool[1], Eva L. Unger[3,4], Andrea R. Bowring[5], Michael D. McGehee[6], Michael F. Toney [1] & Christopher J. Tassone[1]

Understanding the formation chemistry of metal halide perovskites is key to optimizing processing conditions and realizing enhanced optoelectronic properties. Here, we reveal the structure of the crystalline precursor in the formation of methylammonium lead iodide (MAPbI$_3$) from the single-step deposition of lead chloride and three equivalents of methylammonium iodide (PbCl$_2$ + 3MAI) (MA = CH$_3$NH$_3$). The as-spun film consists of crystalline MA$_2$PbI$_3$Cl, which is composed of one-dimensional chains of lead halide octahedra, coexisting with disordered MACl. We show that the transformation of precursor into perovskite is not favored in the presence of MACl, and thus the gradual evaporation of MACl acts as a self-regulating mechanism to slow the conversion. We propose the stable precursor phase enables dense film coverage and the slow transformation may lead to improved crystal quality. This enhanced chemical understanding is paramount for the rational control of film deposition and the fabrication of superior optoelectronic devices.

[1] SSRL Materials Science Division, SLAC National Accelerator Laboratory, 2575 Sand Hill Rd MS 69, Menlo Park, California 94025, USA. [2] Department of Chemistry, Stanford University, Stanford, California 94305, USA. [3] Division of Chemical Physics, Lund University, Lund SE-22100, Sweden. [4] Hybrid Materials Formation and Scaling, Helmholtz-Zentrum Berlin für Materialien und Energie GmbH, Helmholtz-Zentrum Berlin, Albert Einstein Strasse 16, D-12489 Berlin, Germany. [5] Department of Materials Science and Engineering, Stanford University, Stanford, California 94305, USA. [6] Department of Chemical and Biological Engineering, University of Colorado 596 University of Colorado, Boulder, Colorado 80309 - 0596, USA. These authors contributed equally: Kevin H. Stone, Aryeh Gold-Parker. Correspondence and requests for materials should be addressed to M.F.T. (email: mftoney@slac.stanford.edu) or to C.J.T. (email: tassone@slac.stanford.edu)

Metal-halide perovskites have demonstrated impressive optoelectronic properties across a wide compositional space and a range of band gap energies[1]. Surging in popularity since their debut in 2012[2,3], perovskite solar cells have demonstrated a certified efficiency of 22.1%[4] and perovskite materials have shown promise in other optoelectronic applications including photodetectors[5], LEDs[6], and lasers[7]. However, the formation chemistry of perovskite thin films is not always well understood, even in popular deposition routes for the archetypal hybrid perovskite methylammonium lead iodide (MAPbI$_3$, MA = CH$_3$NH$_3$). Across diverse perovskite preparation methods, the conversion chemistry is known to have an impact on film quality, affecting device performance as well as eventual film degradation mechanisms[8–10]. Incomplete understanding of the mechanisms of film formation presently limits the deliberate control of morphology. Improved understanding of the formation chemistry of metal-halide perovskites will guide the development of processing techniques, enabling the rational design of advanced optoelectronic devices.

Early studies of MAPbI$_3$ employed the single-step deposition of equimolar PbI$_2$ and MAI, which were dissolved in the solvent N, N-dimethylformamide (DMF) and spin-cast on a substrate[11,12]. Solar cells made in this manner revealed a non-ideal film morphology: rough, discontinuous films led to low device performance[8]. Other strategies were employed to form smooth, compact films and high-efficiency devices, yet these involved more complicated processes such as sequential deposition[13], vacuum deposition[14], or the use of solution additives[15] or MAI vapor[16]. Large scale production requires well-defined processing conditions and repeatability, but some perovskite deposition routes are poorly understood and irreproducibility can plague even the simplest one-step methods[17]. Thus, achieving morphological control requires a robust understanding of single-step film deposition. This will provide the foundation for studies on multistep deposition routes.

The single-step deposition of MAPbI$_3$ from PbCl$_2$ and 3 stoichiometric units of MAI was found to produce films with dramatically improved morphology and corresponding enhancements in electronic quality compared to films deposited from equimolar PbI$_2$ and MAI[3]. These PbCl$_2$-derived films were shown to have carrier lifetimes of hundreds of nanoseconds[18], among the longest carrier lifetimes measured in perovskite films without additional surface passivation[19]. Distinctively, this deposition does not involve the direct transformation of the starting materials into the final perovskite phase. Rather, the PbCl$_2$-derived processing route forms a crystalline precursor phase immediately after spin casting from the solvent DMF[20]. Through thermal annealing, this precursor undergoes a slow transformation into the nearly chlorine-free MAPbI$_3$[21].

One benefit of PbCl$_2$-derived deposition is that it can achieve full coverage of the substrate, in contrast to single-step deposition from equimolar PbI$_2$ and MAI, which generally produces discontinuous films[22,23]. Improved film coverage may be a general feature of processing methods that employ a crystalline precursor. Seok et al. explained this phenomenon: in single-step processing from equimolar PbI$_2$ and MAI, MAPbI$_3$ crystallizes rapidly during solvent evaporation, leading to the formation of small crystallites and a rough, inhomogeneous film[24]. This can be prevented by depositing a precursor phase that forms in lieu of the perovskite. Following this reasoning, Seok et al. developed a method called solvent engineering, in which the Lewis base dimethylsulfoxide (DMSO) is added to the solvent as a complexing agent. DMSO is known to complex more strongly to lead halides than the solvent DMF[25]. Thus, in solvent engineering approaches, a DMSO-PbI$_2$-MAI[24,25] or DMSO-PbI$_2$[26,27] complex is formed upon spin-coating, which is subsequently converted to a smooth and compact perovskite film. The DMSO complexes have been identified[28,29], and continued research into solvent engineering has led to a record perovskite solar cell efficiency[4]. This success underscores the importance of controlling perovskite formation chemistry in order to direct film morphology and ultimately produce efficient devices.

Unlike in solvent engineering, research suggests that the PbCl$_2$-derived precursor does not incorporate solvent molecules. Munir et al.[17] recorded X-ray diffraction (XRD) during spin-coating and observed no scattering to indicate a precursor-solvate phase. Further evidence was provided by Moore et al.,[30] who recorded identical precursor XRD for PbCl$_2$-derived films deposited from DMF and DMSO solutions. The precursor XRD pattern exhibits some peaks consistent with MAPbCl$_3$, leading some to conclude the presence of the latter in the precursor phase[31–33]. Unger et al.[34] suggested that peaks attributable to MAPbCl$_3$ may instead stem from Pb–Cl bonding distances in a more complex precursor phase, a conclusion shared by Yu et al.[22]. Finally, Pool et al.[21] probed the precursor with Cl K-edge X-ray absorption near edge structure (XANES) measurements, which revealed that the precursor likely contains MACl and an additional species with Cl–Pb bonds.

Another notable aspect of the PbCl$_2$-derived method is the slow formation of the perovskite, requiring a dramatically longer annealing time than is needed after the deposition of equimolar PbI$_2$ and MAI[22,34]. Additionally, PbCl$_2$-derived films are observed to have larger crystallites relative to the chloride-free preparation, as well as reduced microstrain, consistent with a lower density of crystal defects[35]. Slow crystal growth is correlated with improved crystallinity and it has been suggested that the slow formation of PbCl$_2$-derived MAPbI$_3$ leads to higher-quality crystalline domains[22], potentially contributing to the long carrier lifetimes described above.

While the exact chemistry of the transformation has remained elusive, many reports have stressed the importance of MACl evaporation from the film. Moore et al.[30] studied the formation of perovskites made from different lead salts and determined that, for those made from PbCl$_2$, the removal of MACl from the precursor is the major determinant of the conversion kinetics. Although MACl powder sublimes at 190 °C[36], Unger et al.[34] demonstrated that MACl evolves during film annealing at 100 °C, possibly due to a high surface-to-volume ratio in thin films. Finally, Pool et al.[21] demonstrated that XANES spectra show decreasing resemblance to MACl as the film is annealed, further evidence that MACl loss is a significant factor in the conversion process.

The existence of a crystalline precursor and the slow transformation into perovskite are key to the optoelectronic performance of PbCl$_2$-derived perovskites. Yet, the lack of a crystal structure for the precursor has hindered attempts to illuminate the precise chemistry of perovskite formation[30]. In this work, we determine the crystal structure of the precursor phase and measure the evolution of the film structure and chemistry while annealing, which together inform a model that explains why the transformation to perovskite proceeds slowly. In addition, we demonstrate a suite of synchrotron X-ray techniques that can serve as a toolkit to study other methods of perovskite deposition.

## Results

**Precursor structure and characterization.** In a nitrogen glovebox, precursor solutions were prepared by dissolving 0.88 M PbCl$_2$ and 2.64 M MAI in DMF, using typical concentrations that have demonstrated high-quality films[18] and devices[35]. Films were then spin-cast on clean glass substrates as described in Methods. The PbCl$_2$-derived precursor was characterized using grazing

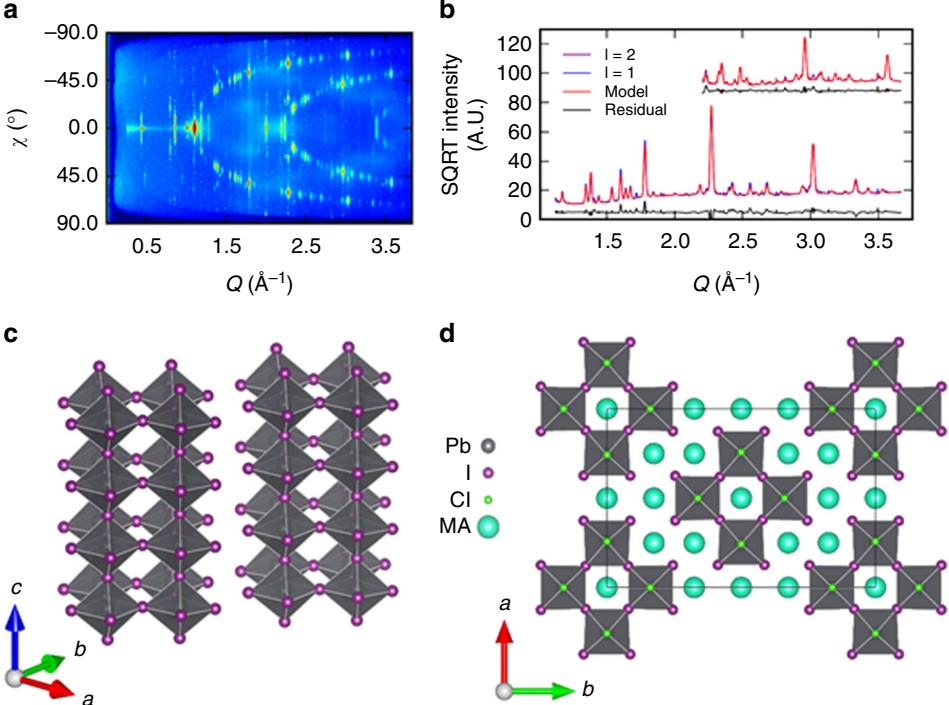

**Fig. 1** Refinement of the precursor crystal structure. **a** GIXRD of precursor film plotted as $Q$ (scattering vector) vs. $\chi$ (azimuthal scattering angle). Apparent arcs are layer lines indexed to (hk1) and (hk2) reflections. **b** Refined model and diffraction data for $l = 1$ and $l = 2$ layer lines. **c** Refined precursor framework: 1D chains with layers of 4 corner-sharing lead-halide octahedra, halides are shown only as iodine and MA+ cations are omitted for clarity. **d** Refined precursor unit cell, MA$^+$ cations (cyan spheres) are added to preserve charge balance. Axial halide sites are colored green to emphasize probable Cl occupancy

incidence X-ray diffraction (GIXRD) on spin-cast films during the initial stages of annealing at 100 °C. The precursor film exhibits a high degree of fiber texture (crystallographic orientation), which leads to discrete diffraction spots instead of powder diffraction rings (Fig. 1a). These spots are organized into clear layer lines that allowed for the indexing of the diffraction pattern. The layer lines correspond to crystalline order along the $c^*$-axis, meaning that the first layer-line is composed of (h,k,1) peaks, the second layer-line composed of (h,k,2) peaks, and so on. Working from this assumption, we were able to index the peaks within the layer lines to a $c$-centered orthorhombic unit cell with dimensions $a = 17.816(3)$ Å, $b = 26.762(6)$ Å, and $c = 5.727(1)$ Å. Any peaks falling outside of the layer lines could also be indexed to this lattice, indicating a secondary orientation of the single precursor phase, described further in Supplementary Note 1.

Isolation of the diffraction from the different layer lines provides nearly single crystal-quality diffraction data, albeit for a select number of Bragg peaks. This was sufficient to solve the precursor structure framework (Fig. 1b). The framework consists of four corner-sharing lead-halide octahedra ordered into extended chains along the $c$-axis, with stoichiometry PbX$_4$ (Fig. 1c). We expect some disorder in the halide sites, as halide ions and vacancies are mobile in the perovskite lattice[37,38]. In addition, the precursor unit cell dimensions are seen to change by as much as 1% while annealing, suggesting that halide occupancies (I vs. Cl) may be dynamic (see Supplementary Fig. 1). Refinement of the halide occupancies suggests a 3:1 ratio of I:Cl with Cl preferentially located in the shared halide sites. In addition, the best fit structure has distinct Pb–X bond lengths. The refinement yields an equatorial bond length of 3.11 Å and two different axial bond lengths: one matching the equatorial length, and one much shorter at 2.89 Å. The shorter of these axial bond lengths determines the $c$-axis lattice parameter. For

comparison, the reported structures of MAPbI$_3$ and MAPbCl$_3$ have Pb–I and Pb–Cl bond lengths of 3.16 Å and 2.84 Å, respectively[39].

Although the diffraction data is insufficient to determine the positions of the MA$^+$ cations, we suggest that these cations reside within the cavities of the chains as well as between chains, which provides for charge balance in the lattice (Fig. 1d). While we cannot rule out the presence of disordered solvent within the film, this model does not leave space for solvent molecules within the crystal structure of the precursor. Combined, the structural refinement and the requirement of charge balance suggest a precursor composition of MA$_2$PbI$_3$Cl.

To further probe the coordination of the lead halide octahedra, we performed extended X-ray absorption fine structure (EXAFS) measurements of the precursor film at the Pb L3-edge. These measurements are sensitive to the local chemical environment of Pb atoms in the film. We fit first-shell EXAFS of the precursor, testing I and Cl at the four equatorial and two (shorter) axial sites (Fig. 2). Allowing the occupancies to vary with bond lengths fixed, the best fit suggests that the equatorial sites (3.11 Å Pb–X bond length) are populated exclusively by I, whereas the axial site (2.89 Å) is composed of 91% Cl and 9% I, giving an overall composition of PbI$_{3.09}$Cl$_{0.91}$ (Fig. 2c). We also fit the EXAFS data for the structures of fixed composition PbI$_4$ (I in equatorial and axial sites) and PbI$_3$Cl (I in equatorial, Cl in axial) (Fig. 2a, b). While the PbI$_4$ model gives a poor fit, the PbI$_3$Cl model is similar to that of PbI$_{3.09}$Cl$_{0.91}$. These fits provide further evidence for chlorine incorporation into the precursor structure with an octahedral stoichiometry similar to PbI$_3$Cl. (More details on EXAFS data processing and fitting are included in Supplementary Methods.) With the above evidence from GIXRD and EXAFS, we identify the crystalline precursor as MA$_2$PbI$_3$Cl and acknowledge some uncertainty regarding the exact halide stoichiometry.

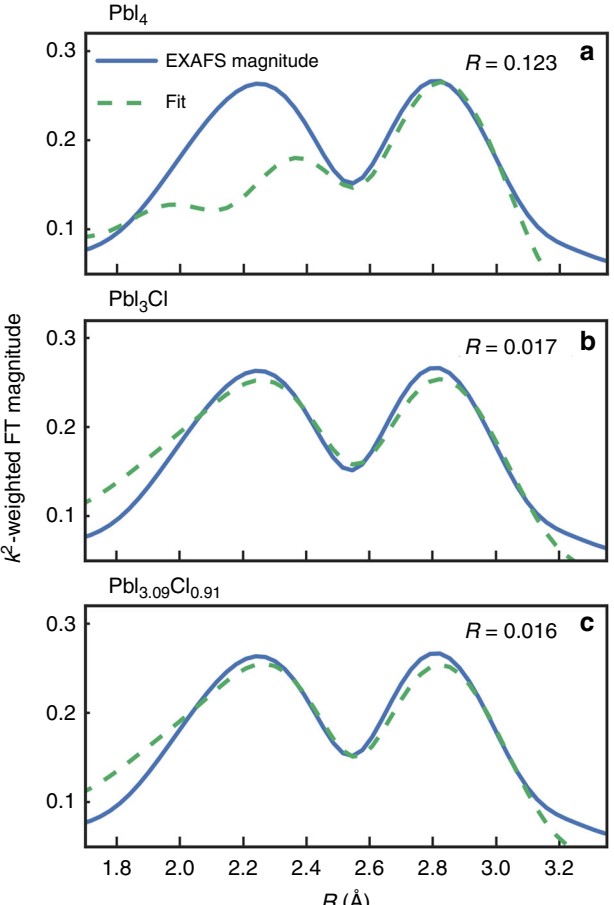

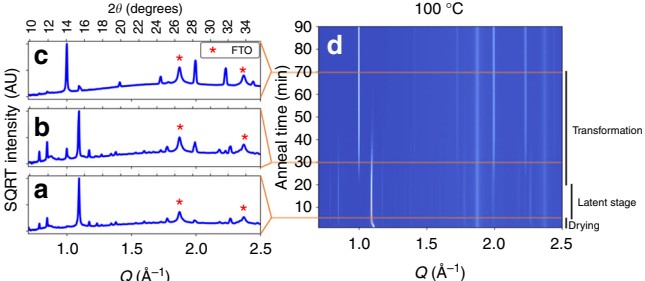

**Fig. 3** In-situ X-ray diffraction of perovskite annealing at 100 °C. **a–c** XRD scans while annealing showing square root intensity to enhance small peaks. XRD scans shown are **a** 5 min, the end of drying, **b** 30 min, during the transformation, and **c** 70 min, near the end of the transformation. **d** Waterfall composite plot: each horizontal slice corresponds to an XRD pattern at a certain time

**Fig. 2** Pb-L$_3$ EXAFS of the precursor film. The fourier transform of the k-space EXAFS is plotted, weighted by $k^2$. Solid lines are magnitude of the measured EXAFS, dashed lines are the fit. $R$ represents misfit; lower is better. **a** Fit to I in all sites (equitorial and axial) of the precursor structure. **b** I in equitorial and Cl in axial sites. **c** Best fit has I in equitorial sites and in 9% of axial sites (rest is Cl)

Given the precursor solution stoichiometry, there must be another molar unit of MACl in the film. We do not observe diffraction peaks from MACl and infer that excess MACl is present in the film as a disordered phase. Thus, we propose that the as-deposited film contains equal fractions of crystalline MA$_2$PbI$_3$Cl (Fig. 1) and disordered MACl.

**Chemistry of transformation from precursor to perovskite.** To study the phase progression from the precursor to the perovskite, we measured in-situ XRD while annealing at temperatures from 95 to 105 °C. Films were deposited by spin-coating in a nitrogen-filled glovebox and transferred to the beamline with no exposure to air. A GIXRD scan was recorded every 20 s, allowing near-continuous observation of the crystal structure transformation. Figure 3 illustrates the structural transformation at 100 °C. The first five minutes show the crystallization of the precursor phase (with characteristic diffraction peaks at $Q = 0.85$, 0.90, 1.1 Å$^{-1}$) as the film dries. Prior studies have observed a disordered precursor solvate in the as-deposited film[17], so this film drying stage likely corresponds to the evaporation of residual DMF. We observe no scattering from crystalline solvate phases, suggesting that the solvent-free crystalline precursor that we observe (Fig. 1) may be more stable than DMF-Pb complexes observed in other deposition strategies. From 5 to 20 min, the precursor scattering stays nearly constant and no perovskite is seen to form ($Q = 1.0$

Å$^{-1}$)—we refer to this as the latent stage. From 20 to 70 min, we observe a continued loss of precursor coincident with the growth of the perovskite phase—this is the transformation of precursor into perovskite. Finally, with no remaining scattering observed from the precursor, the perovskite scattering remains constant until the end of the 90-min annealing.

Once the perovskite begins to form, the transformation proceeds gradually. Using a model-angostic method developed by Mittemeijer and coworkers[40], we measure the time between 20 and 90% of perovskite peak formation as a function of annealing temperature and extract an activation energy of 87 ± 15 kJ/mol (error is standard error in linear fit—see Supplementary Fig. 6 for details). This is in line with similar analysis by Moore et al. (86.6 kJ/mol)[30] and Barrows et al. (85 kJ/mol)[41]. However, this analysis ignores the latent stage, as it only considers the time in which the perovskite is actively forming. The precursor phase is surprisingly persistent: at 100 °C the latent stage lasts a full 15 min. During this time, precursor is not converted into perovskite and the only change observed in the scattering is in the texture of the precursor (see Supplementary Note 1). This latent stage alone is longer than the annealing time to convert a DMSO-perovskite complex into perovskite, which requires only 10 min at 100 °C[24].

Pool et al.[21] showed that Cl evolves from the film continuously throughout annealing, and thus the film chemistry must be changing during the latent stage. We suspected that understanding the evolution of MACl from the film may help explain the unusual latency of this transformation. In order to quantify the evolution of MACl through the different stages of the transformation, we performed in-situ X-ray fluorescence (XRF) on identically prepared films annealed at 95, 100 and 105 °C. XRF allows quantification of the relative amount of Cl in the film while annealing (see Supplementary Figs. 4,5 for details). In Fig. 4, we plot the amount of Cl in the film overlaid with the amount of crystalline perovskite, from integrated XRD peak intensity, both as a function of time. These plots are normalized with respect to the largest amount of Cl measured in the film (at $t = 0$) and to the largest measured intensity of the perovskite diffraction peak. We also plot a linear fit to the first 50% of Cl loss at each temperature. From these fits we calculate an activation energy of 79 ± 32 kJ/mol for the evaporation of MACl (see Supplementary Fig. 7), slightly lower than the 87 kJ/mol for the perovskite conversion, although with large error bars due to the small temperature range investigated.

Our XRF measurements reveal two key insights about the chemical transformation from precursor to perovskite. First: for all temperatures measured, the kinetics of Cl evolution from the film appear remarkably zero-order, or linear, for the majority of

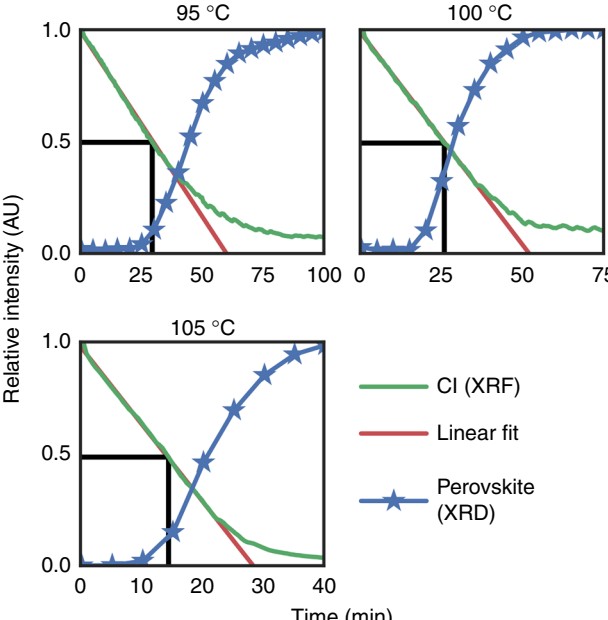

**Fig. 4** Relative amount of Cl (from XRF) and crystalline perovskite (from XRD) while annealing at 95, 100, and 105 °C. Red line is a fit to the linear regime of the XRF, demonstrating the zero-order kinetics of Cl evolution Black lines depict time when half of the Cl has evolved from the film

Cl loss (until 60–80% of the initial Cl has left the film). Our observation fits with the prior report of zero-order kinetics in mass loss from the precursor film[32]. This result is striking, as it suggests that the kinetics of MACl evolution are not limited by the chemical transformation of precursor into perovskite. Rather, zero-order kinetics imply that the rate determining step in the evolution of MACl is desorption from a constant surface area, apparently from the disordered MACl in the film.

The second insight was revealed by comparing the relative amount of Cl in the film with the relative amount of crystalline perovskite. At all three temperatures measured, we find that the onset of perovskite crystallization coincides with 50% of the Cl having evolved from the film, as illustrated with black lines in Fig. 4. Recalling that half of the Cl in the as-deposited film is disordered, this evidence indicates that the transformation is latent until the disordered MACl phase has evaporated. Importantly, this reveals that equilibrium does not favor the forward reaction of precursor to perovskite if any quantity of MACl exists in the film.

These observations suggest a self-regulating model for the transformation, depicted in Fig. 5, in which the conversion of precursor to perovskite is governed by the gradual evaporation of MACl. The exact location of excess MACl within the film cannot be determined, so we postulate excess MACl to exist as a disordered phase at the film surface and/or among the crystallites (Fig. 5a). As illustrated in Fig. 5a, the MACl evaporates from the film during the latent stage. Once this reservoir of MACl is exhausted, the transformation begins, and the crystalline precursor $MA_2PbI_3Cl$ gradually converts into $MAPbI_3$ by loss of the second equivalent of MACl. A chemical model, shown in Fig. 5b, explains the persistent latent stage and the slow rate of transformation. At first, the direct transformation of precursor to perovskite does not proceed because equilibrium does not favor this reaction in the presence of MACl. Instead, the MACl must first evaporate, allowing equilibrium to shift and permitting the forward reaction of $MA_2PbI_3Cl$ to $MAPbI_3$ and MACl. From here the transformation proceeds in a self-regulating manner: as the

conversion advances, more MACl is created, again slowing the conversion until this new MACl has evaporated.

We have demonstrated that the deposition of $MAPbI_3$ from $PbCl_2 + 3MAI$ proceeds via the crystalline precursor $MA_2PbI_3Cl$ and that the transformation to perovskite is governed by the evaporation of MACl. This particular combination of precursor and rate-limiting-step creates a self-regulating mechanism, leading to slow formation of the perovskite. Thus, the $PbCl_2$-derived processing method imparts two distinct benefits to the perovskite film. First, the deposition of a stable precursor prevents rapid perovskite crystallization upon solvent drying, allowing the formation of a compact film without voids or pinholes. Second, the self-regulating release of MACl dramatically slows the transformation of precursor to perovskite, which may lead to high crystalline quality and long carrier lifetimes. We suggest that other precursor-mediated routes could also benefit from a slowed transformation. This could be deliberately engineered: for instance, in solvent engineering methods, the conversion of a $PbI_2$-MAI-DMSO complex could be slowed under a high vapor pressure of DMSO, perhaps producing crystallites of higher quality.

Our in-depth understanding of the precursor chemistry and the chemistry of perovskite formation is enabled here by the combination of XRD and EXAFS to characterize the precursor, and in-situ XRD and XRF to probe the transformation. With these methods, we demonstrate a powerful toolkit for studying perovskite deposition routes to uncover their formation mechanisms. Our improved understanding of $PbCl_2$-derived $MAPbI_3$, both of the crystalline precursor and its transformation into perovskite, represents a step toward the rational control of perovskite film deposition for the next generation of perovskite optoelectronic devices.

## Methods

**Materials**. All chemicals were commercially sourced and used as received without further purification. DMF: Sigma Aldrich Sure/Seal. $PbCl_2$. Aldrich 203572 (99.999% purity). MAI: Dyesol 101000.

**Substrate preparation**. Glass substrates were used for GIXRD, leading to refinement of the precursor. Other films were deposited on FTO substrates with a compact layer of $TiO_2$ in order to resemble the conditions used in n-i-p perovskite devices. Glass and Pilkington TEC15 FTO glass were cleaned by sonication in a diluted Extran solution (EMD, EX0996-1), acetone (EMD, AX0115-1), and iso-propanol (EMD, PX1835P-4). 0.02 M aqueous $TiCl_4$ was spin-coated on the FTO substrates, which were then dried at 75 °C for 15 min, then sintered at 482 °C to convert the layer to $TiO_2$.

**Film deposition**. The 0.88 M $PbCl_2$ and 2.64 M MAI were dissolved in DMF in a $N_2$ glovebox. Solutions were stirred at 75 °C for at least 1 h, until fully dissolved. Films were spin-coated at 2000 rpm for 45 s with solutions and substrates at room temperature.

**Grazing incidence X-ray diffraction**. XRD data was collected at beam line 11-3 of the Stanford Synchrotron Radiation Lightsource. Two-dimensional scattering was collected with monochromatic 12.7 keV X-Rays at an incidence angle of 3° and recorded on a MAR345 Image Plate detector with 345 mm diameter. Samples were measured in a chamber filled with helium that was loaded inside a nitrogen glovebox to avoid exposure to air. $LaB_6$ was used as a calibrant. Images were calibrated and integrated using GSAS-II[42]. Precursor structure solution was performed using TOPAS-Academic through a combination of charge flipping and simulated annealing approaches[43]. Sequential refinement of the structural evolution during annealing was performed using TOPAS-Academic. For more details, see Supplementary Methods.

**Extended X-ray absorption fine structure**. EXAFS measurements were performed on SSRL beamline 4-1 at the Pb-$L_3$ edge. A PIPS solid state detector was used to acquire fluorescence signal, which is the data used herein. Samples were measured in He. EXAFS data were processed using the Demeter package[44]. First, the pre-edge background was subtracted. Then, the average absorption above the edge was removed by fitting to a spline function (Rbkg 1.45). The extracted

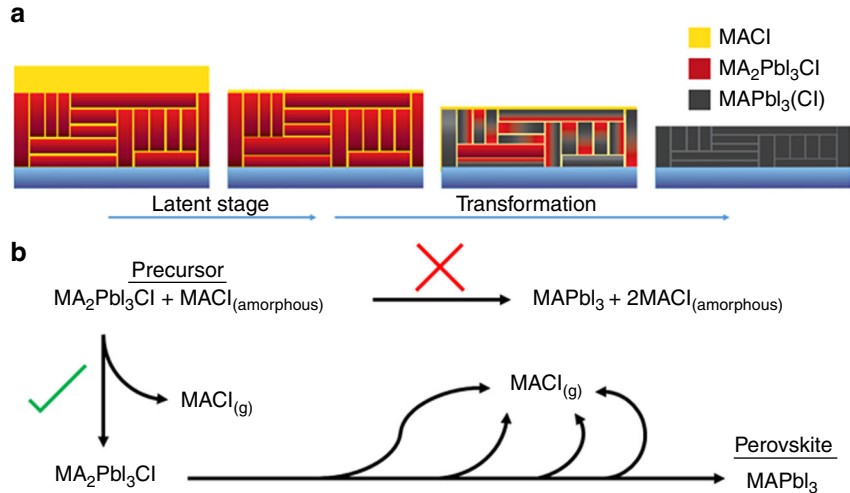

**Fig. 5** Self-regulating model of the transformation from precursor to perovskite. **a** Model detailing the loss of disordered MACl in the latent stage, followed by the conversion of precursor to perovskite with continued loss of MACl regulating the transformation. **b** Chemical model of the transformation from precursor to perovskite. The reaction does not proceed initially in the presence of MACl. Once the existing MACl has evaporated, the transformation can begin, with continued regulation by the newly-formed MACl

absorption $\chi(E)$ was Fourier transformed to real space with a k-window between 3.31 and 12.45 Å$^{-1}$ with window rounding of 1 Å$^{-1}$.

Real-space data was fit between 1.70–3.35 Å using scattering paths for Pb–I and Pb–Cl with distances corresponding to the axial and equatorial sites in the precursor structure. The amplitude reduction factor $S_0^2$ was fixed at 0.85, as determined by EXAFS fitting of a PbI$_2$ standard measured in the same configuration. For more details, see Supplementary Methods.

**In-situ grazing incidence XRD.** 2D GIXRD data were collected as above, except on a Rayonix MX-225 detector measuring 225 × 225 mm$^2$, which allows better time resolution than the MAR345. Scans were integrated to 1D using the Nika software package[45]. Scans were collected every 15 s, and every 20 scans averaged to give 5-min time resolution with improved signal-to-noise. The perovskite cubic (100) peak at $Q = 1.0$ Å$^{-1}$ was isolated in a $Q$-range from 0.95–1.05 Å$^{-1}$ and fit to a line plus a Voigt profile. The integrated intensity of the Voigt is plotted in Fig. 4 as "perovskite (XRD)". For more details, see Supplementary Fig. 3.

**In-situ XRF.** XRF was performed at beamline 4-3. The film was illuminated with 3 keV X-rays and energy-resolved fluorescence was collected with a Vortex detector. The as-spun film was measured in He without air exposure. As the film was annealed, fluorescence was collected continuously and binned every 30 s, showing clear peaks for Pb and Cl. The fluorescence peaks were fit to Gaussians with a constant background, where the integrated intensity of the Gaussian reflects the relative amount of that element in the film. The illuminated volume increases as Cl leaves the film—to account for this, the Cl intensity was normalized by the Pb intensity. This quantity ($I_{Cl}/I_{Pb}$) is plotted in Fig. 4 as "Cl (XRF)." For more details, see Supplementary Methods.

**Data availability.** All relevant data are available by request from the authors.

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

## Acknowledgements

A.G. is supported by NSF GRFP (DGE-1147470). E.U. thanks the Marcus and Amalia Wallenberg Foundation for financial support. V.L.P. was supported by the Department of Energy through the Bay Area Photovoltaic Consortium under Award Number DE-EE0004946. Use of the Stanford Synchrotron Radiation Lightsource, SLAC National Accelerator Laboratory, is supported by the U.S. Department of Energy, Office of Science, Office of Basic Energy Sciences under Contract No. DE-AC02-76SF00515. We thank Ryan Davis, Tim Dunn, Matthew Latimer, and Erik Nelson for help with data collection at SSRL. Doug Van Campen and Valery Borzenets designed the in-situ annealing chambers. A.G. thanks Stanford professors Chris Chidsey and Paul McIntyre for helpful discussions, and Sam Webb and Adam Hoffman for help with EXAFS fitting.

## Author contributions

M.T. and C.T. directed the research. E.U., A.B., and M.M. devised the initial project. E.U. and A.B. prepared samples for GIXRD of the precursor, which E.U., K.S., and C.T. then measured. K.S. refined the precursor structure from the GIXRD data. A.G. and V.P. performed precursor EXAFS and in-situ XRD and XRF of the annealing. A.G. fit the EXAFS and analyzed the in-situ data. K.S., A.G., V.P., M.T., and C.T. developed the model. A.G. wrote the manuscript and all authors contributed edits.

## Additional information

**Competing interests:** The authors declare no competing interests.

