## [Peer Review File · Nature Communications]

Editorial Note: Parts of this peer review file have been redacted as indicated to remove third-party material where no permission to publish could be obtained.

Reviewers' comments:

Reviewer #1 (Remarks to the Author):

In this work, the perovskite crystallization progress from crystalline precursor MA₂PbI₃Cl to final perovskite MAPbI₃ was carefully conducted. The authors proposed that this transforming process is governed by the evaporation of MAI.

1. However, the authors totally ignored the DMF residue in the precursor perovskite film spinning coating from the precursor solution (PbCl₂+ 3MAI+DMF). Previous literatures (Adv. Mater. 29, 1604113 (2017); J. AM. Chem. Soc 137,8696-8699 (2015)) have claimed that the DMF and PbX₂ or MAPbX₃ could form Lewis base adduct in the precursor film. The DMF removal process should also influence the crystallization and morphology formation of perovskite film during thermal annealing. Please add more discussion.
2. The proposed chemical process of figure 4 in this study is not inconsistent with previous reports (Sci. Rep. 5, 10557 (2015)), Wang and his colleagues observed that PbI₂ was produced during the chemical reaction of PbCl₂ and MAI. Please comment.
3. The different heating temperature, or the different MAI evaporation speed could influence the final crystallinity and morphology of drying MAPbI₃ film. Morphology characterizations like SEM or AFM should be added.
4. Line 20-22: can the authors clearly state what kind of formation chemistry is not well understood? There are a lot of work addressing perovskite formation chemistry, only the measuring methods are not these X-ray techniques.
5. Line 34-35: The statement that perovskite formation is poorly understood will mislead this community. The authors should clearly state which part is poorly understood. Also, several groups have reported perovskite devices that are highly reproducible.
6. It seems that the most important motivation of this research is to understand the perovskite formation chemistry. The authors select the PbCl₂ and MAI system, because its film formation is better than PbI₂ and MAI system. Then, later, the authors acknowledge that PbI₂ and MAI system can also form nice films through solvent selection. Several groups have shown that the mixture halide ions in perovskite films is not good for device long term stability. Thus, the PbI₂ and MAI system is much more attractive than the mixed halide precursors. So I suggest the authors give some understanding on perovskite formation chemistry from the PbI₂ and MAI system.
7. In Figure 1c, the readers cannot see the halide anions and organic cations.
8. The Figures are poorly organized and the authors should organize their manuscript carefully. There are two Figure 2's and I could not find Figure 5 (Line 222).
9. The most important issue: there are no electrical measurements on their perovskite films. How do the authors know that their perovskite quality are comparable to those in the literature, since the authors want to show the community the perovskite formation chemistry in previous work.

In view of the above comments and previously published work, the present study does not provide the novelty and scientific rigor required for publication in Nature Communications, and the authors should consider submitting their work to an alternative journal.

Reviewer #2 (Remarks to the Author):

The manuscript describes chemistry of crystallization of MAPbI₃ from the precursors of PbCl₂ and three fold excess MAI. The authors discovered that one dimensional chain structure of MA₂PbI₃Cl with disordered MAI. They found that MAI inhibits transformation of intermediate to perovskite. This findings are very important to understand crystallization of MAPbI₃ from PbCl₂ + 3MAI. Thus this work is highly suitable for Nat. Communications. However, some issues are recommended to be addressed.

1. It is helpful if authors explain the basis for the preferred orientation (or high crystallinity) of resulting MAPbI₃ film especially prepared from PbCl₂ and MAI.

2. According to recent results, perovskite film quality depends on precursor concentration and precursor temperature. The authors used 0.88 M PbCl₂, which was originally proposed by Snaith et al, which however a little bit low concentration. Thus it is required to show the effect of higher concentration of PbCl₂.

3. Authors are recommended to appreciate the pioneering work on the solid state perovskite solar cell reported first in Scientific Reports in 2012, by describing breakthrough in the introduction part, because this work contributed perovskite photovoltaics significantly.

REVIEWER #1

We thank Reviewer #1 for their constructive feedback. Here, we address the comments individually.

1. *The authors totally ignored the DMF residue in the precursor perovskite film spinning coating from the precursor solution (PbCl₂+ 3MAI+DMF). Previous literatures (Adv. Mater. 29, 1604113 (2017); J. AM. Chem. Soc 137,8696-8699 (2015)) have claimed that the DMF and PbX₂ or MAPBX₃ could form Lewis base adduct in the precursor film. The DMF removal process should also influence the crystallization and morphology formation of perovskite film during thermal annealing. Please add more discussion.*

In the PbCl₂-derived preparation, no crystalline DMF-Pb complexes are observed once the film is deposited. Numerous studies have presented X-ray scattering of the precursor phase with no observations of a crystalline solvent complex. We may speculate that the crystalline precursor we report in our manuscript is more stable than DMF-Pb complexes that form in other deposition strategies.

We are familiar with both references suggested by the reviewer and both are cited in the original manuscript. The first reference uses in-situ Grazing Incidence X-ray Diffraction during spin-coating. The authors study numerous preparations, and for the PbCl₂-derived prep they report “a disordered precursor solvate in the as-cast film” where the solvent exists in a “in a more volatile or dynamic state” than in the equimolar preparations. This aligns with our observation of film drying during the first minutes of annealing, leading to a DMF-free crystalline precursor. The majority of the film formation chemistry is then governed by the slow conversion of this precursor into perovskite as we describe at length.

We have added the following clarification to the manuscript, page 5, 2nd full paragraph: “We observe no scattering from crystalline solvate phases, suggesting that the solvent-free crystalline precursor that we observe (Figure 1) may be more stable than DMF-Pb complexes observed in other deposition strategies (Munir et al., 2017).”

The second reference mentioned by the reviewer discusses the stability of lead halide DMSO complexes relative to their DMF counterparts. We discuss these complexes in paragraph 4 of the introduction. We have added the following sentence to page 2, 1st full paragraph: “DMSO is known to complex more strongly to lead halides than the solvent DMF (Ahn et al., 2015).”

2. *The proposed chemical process of figure 4 in this study is not inconsistent with previous reports (Sci. Rep. 5, 10557 (2015)), Wang and his colleagues observed that Pbl₂ was produced during the chemical reaction of PbCl₂ and MAI. Please comment.*

In this reference, the authors observe scattering from Pbl₂ in the as-deposited film, reportedly following the same preparation that we do. They conclude this is a result of the reaction of PbCl₂ and MAI. In contrast, we have not observed scattering from Pbl₂ in repeated measurements of as-deposited films. We only observe scattering from Pbl₂ after degrading the films by annealing for many hours. We ascribe this to the eventual degradation of MAPbl₃, evolving MAI vapor and leaving behind Pbl₂.

Numerous other groups have measured X-ray diffraction of the as-deposited film without air exposure and do not observe scattering from Pbl₂. Specifically, we offer the following citations from our

manuscript as examples: (Moore et al., 2015; Nenon et al., 2016; Tan et al., 2014; Unger et al., 2014). However, the authors of (Tan et al., 2014) do observe PbI_2 in their as-deposited film when measured in air. We have also observed that air exposure will dramatically impact the reaction coordinate. While we cannot say for sure, it is possible that Wang et al. witnessed PbI_2 in their as-deposited film as a result of air exposure. Certainly, their result appears to be an outlier.

- 3. The different heating temperature, or the different MAcl evaporation speed could influence the final crystallinity and morphology of drying MAPbI₃ film. Morphology characterizations like SEM or AFM should be added.*

We agree with the reviewer's comment that different temperatures could influence the final film morphology. In fact, we demonstrate that very small changes in temperature will dramatically affect the reaction kinetics, so it is quite likely that variations in temperature may result in changes in morphology.

We primarily study the transformation process at 100°C as this is the most common method employed in literature. We then present measurements of films annealed at 95 and 105°C to provide evidence for model of the chemical transformation from precursor to perovskite. We make no claims about the final morphology of the films. Thus, we consider the morphological impacts of annealing temperature to be outside the scope of this paper.

Here, we provide SEM measurements of a film we have prepared at 100°C. We show two different levels of magnification alongside comparable measurements from (Sakai et al., 2017), which reports high-efficiency PbCl_2 -derived perovskite solar cells, and from (Yu et al., 2014), a highly-regarded work on this preparation method. From this figure, it is clear that the morphology of our film closely resembles that of high-quality films and devices from literature.

[Redacted]

4. *Line 20-22: can the authors clearly state what kind of formation chemistry is not well understood? There are a lot of wok addressing perovskite formation chemistry, only the measuring methods are not these X-ray techniques.*

The reviewer makes a great point. While the formation chemistry of the PbCl_2 -derived preparation has not been fully elucidated prior to this work, huge strides have been made describing the formation chemistry of other deposition routes. Throughout the introduction, we describe much of the excellent work that has been done over the past few years studying perovskite formation chemistry.

We have edited our statement on page 1, first full paragraph. We have changed “not well understood” to “not always well understood” and we have added a statement to clarify that we refer specifically to certain deposition strategies.

5. *Line 34-35: The statement that perovskite formation is poorly understood will mislead this community. The authors should clearly state which part is poorly understood. Also, several groups have reported perovskite devices that are highly reproducible.*

Again, the reviewer makes an excellent point. We wish to clarify that only some deposition routes are poorly understood. Regarding reproducibility: anecdotally, every perovskite research group we have spoken with has suffered from issues of reproducibility. However, we have made this statement less broad as well.

On page 1, 2nd full paragraph, we have changed “perovskite formation is poorly understood and notoriously irreproducible, even in the simplest one-step methods.” to “some perovskite deposition routes are poorly understood and irreproducibility can plague even the simplest one-step methods.”

6. *It seems that the most important motivation of this research is to understand the perovskite formation chemistry. The authors select the PbCl_2 and MAI system, because its film formation is better than PbI_2 and MAI system. Then, later, the authors acknowledge that PbI_2 and MAI system can also form nice films through solvent selection. Several groups have shown that the mixture halide ions in perovskite films is not good for device long term stability. Thus, the PbI_2 and MAI system is much more attractive than the mixed halide precursors. So I suggest the authors give some understanding on perovskite formation chemistry from the PbI_2 and MAI system.*

Indeed, solvent engineering approaches have enabled high quality films from equimolar PbI_2 and MAI, as we discuss in paragraph 4 of the introduction. As we describe, solvent engineering methods are widely used and have been studied extensively. In fact, greater understanding of film formation and precursor phases has enabled advances in these deposition strategies as well as the development of a world record perovskite solar cell (Yang et al., 2017).

The formation chemistry of films prepared from $\text{PbCl}_2 + 3\text{MAI}$ has not been described in the same level of detail. This manuscript includes the first publication of the precursor structure, as well as a description of the chemistry of transformation from the precursor into the perovskite. As with solvent engineering, we hope this new understanding will enable future advances in devices made from the PbCl_2 -derived preparation.

Finally, the reviewer is concerned that mixed-halide systems show poor device stability. This critique is not relevant to the PbCl_2 -derived method, as we have demonstrated in prior work that very little Cl remains in the fully annealed films, leaving a final halide composition that is almost entirely iodide (Pool, Gold-Parker, McGehee, & Toney, 2015).

7. In Figure 1c, the readers cannot see the halide anions and organic cations.

We thank the reviewer for this comment, as we agree that the clarity of this figure is important. The halide anions which lie at the octahedral corners are small, but we have printed the manuscript and find them to be visible. The organic cations were intentionally omitted from Figure 1c to make the 2×2 corner sharing motif, which makes up the 1D chains, more clearly visible. Cation positions are shown in Figure 1d using a different vantage point. We now realize that this was not clearly explained in the caption. The caption of Figure 1 has been modified to explain this and to reflect that we simplify this figure (1c) to not distinguish between different halides.

8. The Figures are poorly organized and the authors should organize their manuscript carefully. There are two Figure 2's and I could not find Figure 5 (Line 222).

We were very concerned to read this comment. However, on second look, it appears that this is not the case in the manuscript we submitted. Our submitted manuscript has figures labeled 1,2,3,4,5 appropriately.

9. The most important issue: there are no electrical measurements on their perovskite films. How do the authors know that their perovskite quality are comparable to those in the literature, since the authors want to show the community the perovskite formation chemistry in previous work.

We thank the reviewer for this comment. Here, we provide time-resolved photoluminescence measurements on a film we prepared at 100°C . The measured lifetime of 330 ns is comparable to high-quality PbCl_2 -derived films reported in literature, such as lifetimes of 270 ns reported by (Stranks et al., 2013). This measurement demonstrates the high electronic quality of our films and their relevance to PbCl_2 -derived films and devices described in the literature.

10. In view of the above comments and previously published work, the present study does not provide the novelty and scientific rigor required for publication in Nature Communications, and the authors should consider submitting their work to an alternative journal.

We appreciate all the comments and questions raised by the reviewer, and we believe that we have addressed the concerns raised by this reviewer, resulting in a stronger manuscript. We believe that our responses address the reviewers concerns regarding the scientific rigor of our approach, and interpretation of our observations. Specifically, we have explained our reasoning for excluding solvent from the precursor structure, which is both in-line with prior reports as well as supported by the solution to the Reitveld refinement we report in the manuscript. We have clarified specifically what about the formation chemistry is not understood at present, focusing on identifying the chemical species involved in this multi-step conversion process and their respective equilibria. We have addressed concerns regarding the residual chlorine in the film and referenced prior work performed by us which demonstrates that the amount of chlorine in the final film is negligible.

With respect to the novelty of our work, we would respectfully point out that this is the first study which identifies all the transient species involved in the chemical conversion during thermal annealing for this formation chemistry. This is a significant to the broader community for many reasons:

- 1) Disseminating an approach to characterizing the reaction coordinate experimentally which can be replicated for any of the reported perovskite processing schemes reported to date
- 2) Identification of all chemical species in the transformation process will enable first principle calculation of the reaction coordinate. Indeed one of the points raised regarding the role of solvent in the precursor structure, which has not been reported to our knowledge and could not have been confirmed by any other method. Our work enables direct comparison of the reaction coordinates for these two different transition state species.
- 3) We identify, for the first time, how the evolution of MA₂Cl gas limits the growth kinetics of perovskite crystallites during the chemical conversion. As films derived from the PbCl₂ preparation still exhibit the longest carrier lifetimes reported to date, this insight will enable device and process development groups to develop novel deposition strategies which result in less defective absorber layers increasing device PCEs for the even more complicated absorber layer chemistries which represent the current record efficiency devices.
- 4) We are the first group to solve the structure of the precursor for this preparation. As the reviewer points out in-situ x-ray characterization of the conversion process for this preparation has been reported in several publications. None of those publications were able to solve the precursor structure from their data. This structure solution is important for the reasons mentioned above, but also as it details a method for performing a similar refinement from in-situ x-ray characterization using area detector commonly performed to characterize chemical transformations.

In light of these points, we respectfully disagree that the novelty of this work is lacking, and would ask the reviewer to reconsider in light of our responses and alterations to the manuscript.

REVIEWER #2

Reviewer #2 found our submission “highly suitable for publication in Nature Communications.” We thank Reviewer #2 for their insightful comments – this feedback has helped to further improve this manuscript.

1. It is helpful if authors explain the basis for the preferred orientation (or high crystallinity) of resulting MAPbI₃ film especially prepared from PbCl₂ and MAI.

The reviewer asks a very interesting question about preferred orientation. Curiously, the precursor film is highly oriented, but the final perovskite film is only somewhat oriented, suggesting some degree of dissolution-recrystallization during the conversion process. There are a number of studies wholly concerned with the orientation of perovskite films (Chen et al., 2017; Oesinghaus et al., 2016). While the preferential orientation of the precursor was instrumental in our Rietveld refinement of the structure, we do not have evidence that the orientation is relevant in the conversion process. We discuss crystal orientation in greater detail in the SI.

2. According to recent results, perovskite film quality depends on precursor concentration and precursor temperature. The authors used 0.88 M PbCl₂, which was originally proposed by Snaith et al, which however a little bit low concentration. Thus is is required to show the effect of higher concentration of PbCl₂.

The ratio of PbCl₂:3MAI is key to this conversion process, enabling a crystalline precursor with excess MAI, as we describe in the manuscript. This precursor concentration of 0.88M PbCl₂ has been employed extensively from the early work that popularized this method (Stranks et al., 2013) to the recent best-in-class PbCl₂-derived solar cells (Sakai et al., 2017). To ensure that our results are relevant, we have thus employed this same precursor concentration. While the impact of precursor concentration may be a very interesting study, it is outside the scope of this work.

We have added the phrase, “using typical concentrations that have demonstrated high-quality films (Stranks et al., 2013) and devices (Sakai et al., 2017)” to page 3, 2nd full paragraph.

3. Authors are recommended to appreciate the pioneering work on the solid state perovskite solar cell reported first in Scientific Reports in 2012, by describing breakthrough in the introduction part, because this work contributed perovskite photovoltaics significantly.

We thank the reviewer for the comment. We have added this citation to the introduction.

REFERENCES

- Ahn, N., Son, D.-Y., Jang, I.-H., Kang, S. M., Choi, M., & Park, N.-G. (2015). Highly Reproducible Perovskite Solar Cells with Average Efficiency of 18.3% and Best Efficiency of 19.7% Fabricated via Lewis Base Adduct of Lead(II) Iodide. *Journal of the American Chemical Society*, 137(27), 8696–8699. <http://doi.org/10.1021/jacs.5b04930>
- Chen, A. Z., Foley, B. J., Ma, J. H., Alpert, M. R., Niezgodna, J. S., & Choi, J. J. (2017). Crystallographic orientation propagation in metal halide perovskite thin films. *Journal of Materials Chemistry A*,

5(17), 7796–7800. <http://doi.org/10.1039/C7TA02203D>

- Moore, D. T., Sai, H., Tan, K. W., Smilgies, D., Zhang, W., Snaith, H. J., ... Estroff, L. a. (2015). Crystallization kinetics of organic-inorganic trihalide perovskites and the role of the lead anion in crystal growth. *Journal of the American Chemical Society*, *137*(6), 2350–2358. <http://doi.org/10.1021/ja512117e>
- Munir, R., Sheikh, A. D., Abdelsamie, M., Hu, H., Yu, L., Zhao, K., ... Amassian, A. (2017). Hybrid Perovskite Thin-Film Photovoltaics: In Situ Diagnostics and Importance of the Precursor Solvate Phases. *Advanced Materials*, *29*(2), 1604113. <http://doi.org/10.1002/adma.201604113>
- Neon, D. P., Christians, J. A., Wheeler, L. M., Blackburn, J. L., Sanehira, E. M., Dou, B., ... Luther, J. M. (2016). Structural and chemical evolution of methylammonium lead halide perovskites during thermal processing from solution. *Energy Environ. Sci.*, *9*(6), 2072–2082. <http://doi.org/10.1039/C6EE01047D>
- Oesinghaus, L., Schlipf, J., Giesbrecht, N., Song, L., Hu, Y., Bein, T., ... Müller-Buschbaum, P. (2016). Toward Tailored Film Morphologies: The Origin of Crystal Orientation in Hybrid Perovskite Thin Films. *Advanced Materials Interfaces*, *3*(19), 1600403. <http://doi.org/10.1002/admi.201600403>
- Pool, V. L., Gold-Parker, A., McGehee, M. D., & Toney, M. F. (2015). Chlorine in PbCl₂-Derived Hybrid-Perovskite Solar Absorbers. *Chemistry of Materials*, *27*(21). <http://doi.org/10.1021/acs.chemmater.5b03581>
- Sakai, N., Wang, Z., Burlakov, V. M., Lim, J., McMeekin, D., Pathak, S., & Snaith, H. J. (2017). Controlling Nucleation and Growth of Metal Halide Perovskite Thin Films for High-Efficiency Perovskite Solar Cells. *Small*, *13*(14), 1–8. <http://doi.org/10.1002/smll.201602808>
- Stranks, S. D., Eperon, G. E., Grancini, G., Menelaou, C., Alcocer, M. J. P., Leijtens, T., ... Snaith, H. J. (2013). Electron-Hole Diffusion Lengths Exceeding 1 Micrometer in an Organometal Trihalide Perovskite Absorber. *Science*, *342*(6156), 341–344. <http://doi.org/10.1126/science.1243982>
- Tan, K. W., Moore, D. T., Saliba, M., Sai, H., Estroff, L. a, Hanrath, T., ... Wiesner, U. (2014). Thermally Induced Structural Evolution and Performance of Mesoporous Block Copolymer-Directed Alumina Perovskite Solar Cells. *ACS Nano*, *8*(5), 4730–4739. <http://doi.org/10.1021/nn500526t>
- Unger, E. L., Bowring, A. R., Tassone, C. J., Pool, V. L., Gold-Parker, A., Cheacharoen, R., ... McGehee, M. D. (2014). Chloride in Lead Chloride-Derived Organo-Metal Halides for Perovskite-Absorber Solar Cells. *Chemistry of Materials*, *26*(24), 7158–7165. <http://doi.org/10.1021/cm503828b>
- Yang, W. S., Park, B.-W., Jung, E. H., Jeon, N. J., Kim, Y. C., Lee, D. U., ... Seok, S. Il. (2017). Iodide management in formamidinium-lead-halide-based perovskite layers for efficient solar cells. *Science*, *356*(6345). Retrieved from <http://science.sciencemag.org/content/356/6345/1376>
- Yu, H., Wang, F., Xie, F., Li, W., Chen, J., & Zhao, N. (2014). The Role of Chlorine in the Formation Process of “CH₃NH₃PbI_{3-x}Cl_x” Perovskite. *Advanced Functional Materials*, *24*(45), n/a-n/a. <http://doi.org/10.1002/adfm.201401872>

Reviewers' comments:

Reviewer #1 (Remarks to the Author):

The authors have addressed the technical criticisms that I raised in my original review, but I feel that the impact of the work leaves a lot to be desired and does not, in my opinion, rise to the level of a manuscript in the Nature family and feel that this work is better suited to a journal that focuses on PV. The technical level is absolutely fine, it is more a matter of the venue and I do not see this as being appealing to a broad audience.

The revised manuscript added more discussions based on the reviewer's comments. However, there are still some key questions need to be addressed clearly.

(1) The solvent residue issue for the as coating perovskite film prepared in this study. The DMF residue exists in the precursor perovskite film just after spinning coating without thermal annealing. The authors claimed that (line 64-66 in maintext) "Munir et al. recorded X-ray Diffraction (XRD) during spin-coating and observed no scattering to indicate a precursor-solvate phase¹⁷. Further evidence was provided by Moore et al., who recorded identical precursor XRD for PbCl₂-derived films deposited from DMF and DMSO solutions³⁰". Also, in the rebuttal letter of "In the PbCl₂-derived preparation, no crystalline DMF-Pb complexes are observed once the film is deposited." These are not scientific.

Here, the XRD could be utilized for the perovskite crystallization characterizations, not for the chemical composition information. There are some other techniques could be applied for the chemical composition, and the chemical evolution of the coated perovskite film during annealing process, such as the (in-situ) Fourier transform infrared (FTIR). For the reference 17 (Adv. Mater. 29, 1604113 (2017)) cited in the maintext, the author did observe the DMF-Pb complex in the as-casted film, please see the figure 1f and figure S6 as below. And the authors in reference 17 clearly explained that DMF could be disappeared after thermal annealing. The FTIR of the spin coated film with and without annealing process must be added, to check the DMF residue in this study. If the DMF residue existed, the model proposed in figure 5 must be modified. And in situ FTIR of the perovskite film annealing process should also be added.

[Redacted]

[Redacted]

(2) For the MACl guided perovskite crystallization model shown in figure 5a, the EDS (energy-dispersive x-ray spectroscopy) images of the cross-sectional film samples with different annealing time should be added. The depth profile of Pb element should be changed with different annealing time along with the continued loss of MACl.

Basing on the above discussion, some of the data is not sufficient and some discussions are not scientific. We still not recommend this paper to be published in Nature Communications.

Reviewer #2 (Remarks to the Author):

The manuscript has been revised accordingly by addressing all the queries from the reviewers. Thus the revised version is now suitable for publication.

Reviewer #3 (Remarks to the Author):

I have read and reviewed the paper of Stone et al. and also their response to the referee's and their comments. I find the paper highly interesting and highly relevant and I deem that it should be published in nature Communications. They have finally revealed the precursor phase from the PbCl₂ processing route, which will have significant impact on both the science and technology going forward. I have no technical suggestions for the Authors. I will briefly explain below why I judge this article to be of such interest:

The paper of Stone et al. determines for the first time the crystalline precursor phase for the PbCl₂ derived MAPbI₃. This specific preparation route is highly significant both historically and technologically. It was the processing of the perovskite through this route originally in 2012, which allowed high quality crystalline films to be deposited and the subsequent discovery and development of efficient thin-film perovskite solar cells. Over the last few years, as is well described in the article, this route has been broadened to include other Pb- precursors, including non-halide precursors of Acetate and nitrate, with acetate also being quite broadly employed. What is known to date is that there is a precursor phase formed when the perovskite is crystallised through these routes, but exactly what that precursor is, is unknown, until now.

The importance of understanding precursor phases: It is probably fair to say that it has emerged over the last few years that ALL the most efficient routes to processing perovskite films, go through a precursor phase. These are either the route described here, or the route where a DMSO-PbX complexation precedes perovskite film crystallisation. Therefore there is a present challenge of working out what is the ideal perovskite precursor phase to crystallise through, to deliver the best quality perovskite. A Second challenge of how do we best control the crystallisation through the precursor phase. However, without knowledge of what these precursor phase are, very little intelligent work can be done, and the community has had to resort to an inefficient trial and error approach. Now, due to this work of Stone et al., for this PbCl₂ and related routes, there is a clear picture of what the precursor phase is. Hence we should now be able to gain better control of both its formation, and the subsequent transformation into the 3D perovskite phase. To mention just a few possibilities, instantly re-optimisation of the solution composition can take place, which will lead directly to the precursor phase without the unnecessary "excess excess" of MAcl, the expansion of this route into other more contemporary compositions, such as mixed anion and mixed metal perovskites, extending this to the 2D and ruddelson popper phases etc etc.

In short, I deem this a fantastic and very inspiring piece of work, which will definitely (re)inspire (re)investigation and advancement of these perovskite processing routes.

This letter constitutes our formal response to the review of our work "PbCl₂ Derived MAPbI₃: Transformation from Crystalline Precursor to Perovskite" by Reviewer #1 following the second round of peer review comments. With respect to the reviewer's comment on including in-situ FTIR measurements during the annealing process, we would respectfully ask the editor not to make inclusion of such a study a condition of acceptance of our submitted paper. First and foremost, we do not anticipate that this will result in additional information which is not already present in the existing literature, or covered by the work reported in our paper.

The FTIR measurement suggested by the reviewer is a nice way to probe a wet film that exhibits little X-ray scattering before the annealing step (Ref 17; our manuscript Figure 3). In-situ FTIR may be able to prove conclusively when DMF leaves the film, however we believe that the prior work of Munir has already demonstrated that a disordered solvated phase exists after spin casting and prior to annealing. Replicating that result in our study would not be a novel result, nor increase the impact of this work. Furthermore given the following evidence, we do not believe that such a study would demonstrate to what extent DMF impacts the transformation of the crystalline precursor to the crystalline perovskite:

1. DMF forms a disordered (non-crystalline) phase in the as-spun film (Ref 17, our figure 3)
2. DMF complexes much less strongly to lead halides than does DMSO (Refs 24, 25)
3. DMF evaporates rapidly (within minutes) in one-step perovskite depositions (Refs 24, 25)
4. The PbCl₂-derived precursor exhibits identical X-ray diffraction whether deposited from DMF or DMSO, indicating no solvent incorporation (Ref 30)
5. From in-situ GIXD, the film is observed to dry within 5 minutes, leading to the crystallization of the precursor phase (our manuscript).

For these reasons, it is clear that the crystalline precursor phase does not contain DMF, and it is extremely likely that the vast majority of DMF evaporates in the first few minutes of annealing. We do agree with Reviewer #1 that the solvent choice will likely impact the kinetics of the transformation, and possibly the thermodynamics as well. However, such a study would require contrasting the effects of many different solvents across preparation methods, which we believe is outside the scope of this work. We briefly discuss this in the 2nd full paragraph of page 5, but are happy to add the highlighted sentence to make this point more clearly.

The first five minutes [of annealing] show the crystallization of the precursor phase (with characteristic diffraction peaks at $Q=0.85, 0.90, 1.1 \text{ \AA}^{-1}$) as the film dries. Prior studies have observed a disordered precursor solvate in the as-spun film (Ref 17), so this film drying stage likely corresponds to the evaporation of residual DMF in the as-deposited film.

Reviewer #1 has also suggested that we perform additional EDX cross-sectional measurements. Although we agree with the reviewer that EDX could ideally provide information on elemental depth distribution, we regret that these measurements are not

likely to be possible without altering the state of the film. A number of EDX measurements are reported on PbCl_2 -derived films, including by our research group (Ref 34). Unfortunately, these measurements are hampered by the necessary measurement conditions. Importantly, vacuum has a significant impact in removing MgCl_2 from these films, especially those that are not fully-converted. It has been demonstrated that PbCl_2 -derived films can be fully converted in only 30 minutes at 60° under vacuum (1 mbar pressure), due to the substantial increase in MgCl_2 evaporation rate (Xie, Feng Xian, et al. "Vacuum-assisted thermal annealing of $\text{CH}_3\text{NH}_3\text{PbI}_3$ for highly stable and efficient perovskite solar cells." *ACS nano* 9.1 (2015): 639-646.). This is consistent with our model of a conversion process limited by MgCl_2 evaporation. In addition, the intense electron beam may alter the sample, potentially resulting an altered elemental distribution. We believe that including such a study would introduce additional variables for which we cannot control, nor would they be representative of the processing of these films which occur at ambient pressure. For this reason, we have endeavored to only employ ambient-pressure measurements in the present manuscript.